# Volatilomics Analysis of Jasmine Tea during Multiple Rounds of Scenting Processes

**DOI:** 10.3390/foods12040812

**Published:** 2023-02-14

**Authors:** Cheng Zhang, Chengzhe Zhou, Caiyun Tian, Kai Xu, Zhongxiong Lai, Yuling Lin, Yuqiong Guo

**Affiliations:** 1College of Horticulture, Fujian Agriculture and Forestry University, Fuzhou 350002, China; 2Tea Industry Research Institute, Fujian Agriculture and Forestry University, Fuzhou 350002, China; 3Institute of Horticultural Biotechnology, Fujian Agriculture and Forestry University, Fuzhou 350002, China

**Keywords:** jasmine tea, scenting process, refreshing aroma, widely targeted volatilomics

## Abstract

Jasmine tea is reprocessed from finished tea by absorbing the floral aroma of jasmine (*Jasminum sambac* (L.) Aiton); this process is commonly known as “scenting”. Making high-quality jasmine tea with a refreshing aroma requires repeated scenting. To date, the detailed volatile organic compounds (VOCs) and the formation of a refreshing aroma as the number of scenting processes increases are largely unknown and therefore need further study. To this end, integrated sensory evaluation, widely targeted volatilomics analysis, multivariate statistical analyses, and odor activity value (OAV) analysis were performed. The results showed that the aroma freshness, concentration, purity, and persistence of jasmine tea gradually intensifies as the number of scenting processes increases, and the last round of scenting process without drying plays a significant role in improving the refreshing aroma. A total of 887 VOCs was detected in jasmine tea samples, and their types and contents increased with the number of scenting processes. In addition, eight VOCs, including ethyl (methylthio)acetate, (Z)-3-hexen-1-ol acetate, (E)-2-hexenal, 2-nonenal, (Z)-3-hexen-1-ol, (6Z)-nonen-1-ol, β-ionone, and benzyl acetate, were identified as key odorants responsible for the refreshing aroma of jasmine tea. This detailed information can expand our understanding of the formation of a refreshing aroma of jasmine tea.

## 1. Introduction

In China, it has been said since antiquity that “the tea attracts flowers’ fragrance to benefit the tea flavor”. The flower-scented teas are produced by mixing base teas such as green, white, black, or oolong teas with flowers so that they naturally absorb the floral aroma [1]. Among them, jasmine tea, which is traditionally produced in Fujian province of China by using the flowers of jasmine (*Jasminum sambac* (L.) Aiton) as scenting material, is the most popular [2]. According to the statistics of the China Tea Mobilization Association (https://www.ctma.com.cn/, accessed on 3 November 2022), in 2020, China’s jasmine tea production reached 112,500 tons, was sold worldwide, and created a gross product of 2.676 billion yuan. The appeal of jasmine tea stems from its unique aroma as well as its sedative effects on the autonomic nerve and mood states [3,4]. High-quality jasmine tea has a refreshing jasmine aroma, and the aroma is pure, concentrated, and long-lasting [5]. Volatile organic compounds (VOCs) are fundamental for the jasmine tea aroma, and many researchers have emphasized the key odorants responsible for jasmine tea [6,7,8]. On the basis of their chemical structure, these key odorants can be mainly classified into terpenoids, esters, aldehydes, alcohols, aromatics, and others (such as indole and methyl anthranilate). Furthermore, these VOCs are also the key odorants of jasmine flower [9,10,11].

As the VOCs of jasmine tea are mainly absorbed from the fresh jasmine flowers by the base tea [1,12], the quality of flowers and base tea is the basis of the jasmine tea aroma, while the scenting technology determines the formation of jasmine tea aroma quality [6,13]. In November 2022, the “Traditional Chinese Tea-making Techniques and Related Practices” was inscribed on the UNESCO Representative List of the Intangible Cultural Heritage of Humanity, including the “Fuzhou Jasmine Tea Scenting Process”. In actual production, the baked green tea produced in spring is the main base tea used for making jasmine tea [14]. However, the production of jasmine tea usually takes place during the summer months because the jasmine flowers bloom with the best aroma quality in summer nights [15]. For the scenting process, the jasmine flowers and base tea are placed in alternate layers, allowing the tea leaves to fully absorb the aroma of the jasmine flowers, and the total time required for this process is about 12 h. Afterwards, flowers that have lost their scent are separated from the tea leaves, and the tea leaves are dried out to conserve their aroma. Because jasmine tea aroma is determined by both the adsorption and retention of VOCs in the processed tea leaves, the scenting process can be repeated several times until the desired tea quality is reached [12]. The last round of scenting is done by mixing tea leaves with a small amount of the flowers for 4–6 h without a drying process to obtain the finished tea. This special scenting process is known as “Tihua” [7], which is necessary to give jasmine tea its refreshing aroma quality. Previous research has shown that different technological parameters of the scenting process could largely affect the aroma quality of jasmine tea [6]. Moreover, Chen et al. [7] reported that the total amount of VOCs increased gradually with increasing scenting rounds. However, the impacts of multiple rounds of scenting processes, especially the “Tihua” process, on the formation of a refreshing aroma of jasmine tea remain largely unknown. 

In the present study, we aimed to investigate the role of repeated scenting processes in the formation of a refreshing aroma of jasmine tea. To this end, sensory evaluation, as well as headspace solid-phase microextraction (HS-SPME) combined with gas chromatography–mass spectrometry (GC–MS)-based widely-targeted volatilomics (WTV) methods were performed to investigate the dynamic changes of VOCs and aroma properties in jasmine teas during multiple rounds of scenting processes. In addition, multivariate statistical analyses were performed to determine the law of the dynamic evolution of VOCs in jasmine tea with different rounds of scenting. This study provides a theoretical reference and objective basis for the formation mechanism of high-quality jasmine tea.

## 2. Materials and Methods

### 2.1. Manufacturing and Collection of Tea Samples

The jasmine tea samples were produced in Fuzhou, China, from August 1st to 17th, 2022. The baked green tea (*Camellia sinensis* (L.) Kuntze cv. Fudingdabaicha) and the jasmine flower (*J. sambac* cv. *bifoliatum*) were used as the base tea and scenting material, respectively. The base tea was produced by Fuding Mingchun Tea Co., Ltd. (Fuding, Fujian, China) in April 2022 and stored at 4 °C. Before the processing of jasmine tea, a part of the base tea was collected as a control check (CK) sample and named BT. The manufacturing procedures are specified in the Chinese national standard procedure (GB/T 34779—2017) and are shown in Figure 1. Briefly, unopened mature flower buds were picked on the sunny afternoon of August 1st and maintained at 35 ± 2 °C. When flowers bloomed to “bowl” shapes at about 9:00 p.m., they were used for the first round of scenting. About 70 kg of flowers were mixed with 100 kg of base tea to form a pile to absorb the flower aroma. During this period, when the temperature of the pile exceeded 45 °C, the pile was spread out for heat dissipation until the temperature dropped to about 35 °C, and then the pile was re-heaped up for scenting. After 12 h of scenting, the flowers were separated from the tea leaves and then were dried at about 110 °C until the moisture content of the tea sample was below 6%, named the R1 sample. On August 2022, about 60 kg of flowers was mixed with 100 kg of the R1 sample for the second round of scenting, and other processing parameters remained the same as the first round of scenting except for adjusting the drying temperature to 100 °C, and the R2 sample was obtained. On 10 August 2022, the third round of scenting was performed by mixing 50 kg of flowers and 100 kg of the R2 sample and maintaining the same processing parameters as the last round of scenting to obtain the R3 sample. On 15 August 2022, we used 40 kg of flowers for scenting the R3 sample with the processing parameters as the second round of scenting, and this yielded the R4 sample. On the evening of 16 August, 100 kg of the R4 sample was mixed with 10 kg of flowers for 6 h for the Tihua process, which yield the T1 sample. Taken together, we immediately obtained the BT, R1, R2, R3, R4, and T1 samples once they were produced. Each sample was composed of three biological replicates, stored at 4 °C, and kept dry for further analysis.

### 2.2. Evaluation of Aroma Quality

A panel of five expert panelists (two males, three females, ranging from 40 to 53 years old) was recruited to evaluate the aroma profiles of different tea samples, including four aroma attributes (freshness, concentration, persistence, and purity of jasmine aroma). The evaluation method referred to the Chinese national standard (GB/T 23776—2018). Briefly, 3 g of the tea sample was brewed with 150 mL of boiled water in a cylindrical cup for 3 min and 5 min, respectively, for the first and second rounds of evaluation. The first round of evaluation mainly estimated the freshness and purity of the jasmine aroma, while the second round of evaluation focused on the concentration and persistence of the jasmine aroma. The panelists rated the intensity of aroma attributes on an eight-point scale, ranging from 0 (absent) to 8 (extremely strong). Before evaluation, the panelists were trained using the same tea samples to standardize attributes. The average score for each attribute was used for the quantitative description of aroma quality.

### 2.3. HS-SPME Procedure and GC–MS Analysis

The collection and identification of VOCs of tea samples were performed using HS-SPME combined with GC–MS, as previously described [9] with minor modifications. Specifically, tea samples were separately ground into powder using an MM400 ball mill (Retsch, Shanghai, China) and then weighed, and 0.50 g of powder sample was added into a 20 mL glass vial containing 2 mL of saturated NaCl (analytical pure) solution. The HS-SPME procedure was performed using an SPME AllowCond instrument (CTC Analytics AG, Zwingen, Switzerland) for collecting VOCs. The glass vials were shaken at 60 °C for 5 min, and then a 120 µm divinylbenzene/carboxen/polydimethylsiloxane SPME Arrow (incubated under a Fiber Conditioning Station at 250 °C for 2 h before use) was inserted into the glass vial to absorb the VOCs for 15 min. Subsequently, the desorption of VOCs was performed in the injection port of the 7890B GC apparatus (Agilent, Palo Alto, CA, USA) at 250 °C for 5 min and then were carried by helium at a constant velocity of 1.2 mL/min in the splitless mode for injection. The DB-5MS capillary column (30 m × 0.25 mm × 0.25 μm) was used for separating VOCs. The column temperature was initially set to 40 °C for 3.5 min, rising to 100 °C at 10 °C/min, 180 °C at 7 °C/min, and 280 °C at 25 °C/min, and then held for 5 min. Afterwards, the 7000D MS (Agilent) was used to identify the VOCs. The mass spectrometer was operated in electron impact (EI) ionization mode with an ionizing energy of 70 eV. The temperatures of the ion source, quadrupole, and transfer line were set at 230, 150, and 280 °C, respectively. The timed selected ion monitoring mode was performed for data acquisition. All powder samples were mixed as quality control (QC) samples and were inserted into every six test samples to monitor the repeatability of the detection results.

### 2.4. WTV Method

The volatilomics based on SPME combined with GC–MS analysis is the most common approach to study tea volatilomes because it can analyze all the detected metabolites at once, including chemical unknowns [16,17,18]. However, the traditional method comes with many shortcomings, such as spectral convolution, low limit of detection, and poor identification coverage and reproducibility [19,20,21]. The WTV method was reported to overcome the defects of the traditional approach described above that uses a “targeted spectra extraction” algorithm for addressing spectral convolution, constructing an MS^2^ spectral tag library to expand chemical annotation, adapting a multiple-reaction-monitoring mode to improve sensitivity, and using regression models to correct for signal drift [22]. For the qualitative and quantitative measurements of VOCs, the WTV method was employed according to a previous report [22] with some adjustments. The self-built library, MWGC, includes determined retention time (RT) as well as qualitative and quantitative ions for precise scanning in selected ion detection mode, with one quantitative ion and two to three qualitative ions selected for each VOC, respectively. Raw data were analyzed using an Agilent MassHunter. All ions to be detected in each group were detected separately in the order of peak emergence, and if the RT was consistent with the standard reference and the selected ions were all present in the mass spectra of the samples after background deduction, the VOC was identified. The 3-hexanone-2,2,4,4-d4 (CAS: 24588-54-3) was used as internal standard to normalize data processing as per a previous report [23].

### 2.5. Multivariate Statistical Analysis Methods

#### 2.5.1. Data Processing of Quantitative Descriptive Analysis

The average score of each aroma attribute was visualized as a radar plot by using Origin software (version 2021, Originlab Corporation, Northampton, MA, USA). Significance differences were determined using SPSS (version 25.0, IBM, Armonk, NY, USA), with one-way analysis of variance (ANOVA) followed by Duncan’s multiple range test. Quantitative analysis data were expressed as the mean ± standard deviation (SD).

#### 2.5.2. Coefficient of Variation

Coefficient of variation (CV) values were obtained by calculating the ratio of the standard deviation of the raw data to the mean of the raw data. The frequency of CV occurrences of metabolites smaller than the reference value was analyzed using the Empirical Cumulative Distribution Function.

#### 2.5.3. Correlation Analysis between Every Two Samples

Pearson correlation coefficients (PCCs) between every two samples were calculated using the built-in cor function of the R package (version 3.5.1). 

#### 2.5.4. Principal Component Analysis

The raw data of identified VOCs, which were Z-score normalized, were subjected to unsupervised principal component analysis (PCA) using the statistics function prcomp within the R package (version 3.5.1).

#### 2.5.5. Orthogonal Partial Least Squares-Discriminant Analysis and Differential VOCs Selected

The raw data of identified VOCs were normalized via log_2_ conversion and then subjected to zero-centering. The processed data were used for orthogonal partial least squares-discriminant analysis (OPLS-DA) using MetaboAnalystR within the R package (version 1.0.1), with the OPLSR.Anal function and 200 permutations. The variable importance in projection (VIP) values were extracted from the OPLS-DA results. Significantly differential VOCs between inter-groups accorded with VIP ≥ 1 and log_2_fold change (FC) ≥ 1. The relationship between the number of differential VOCs in each group was presented as a Venn diagram using TBtools (version 1.1043) [24].

#### 2.5.6. K-Means Clustering Analysis

To investigate the changing trends of the VOCs in different groups, the differential VOCs identified in all the compared comparisons were subjected to Z-score standardization followed by K-means clustering analysis using the R base package (version 3.5.1). 

#### 2.5.7. Partial Least Squares Regression Analysis

Partial least squares regression (PLSR) was performed using SIMCA software (version 14.1) (Umetrics, Umea, Sweden).

### 2.6. Quantification and Calculation of Odor Activity Values

The odor activity values (*OAVs*) of VOCs were calculated using the following formula:(1)$$OAV_{i}\;=\;\frac{C_{i}}{OT_{i}}$$
where *C_i_* (µg kg^−1^) and *OT_i_* (µg kg^−1^) represent the individual VOC content and its aroma threshold in water, respectively. A VOC with an *OAV* > 1 can be perceived by the human nose [9].

## 3. Results and Discussion

### 3.1. Repeated Scenting Processes Improve the Aroma Quality of Jasmine Tea

Aroma is the most important indicator in determining the quality of jasmine tea, and whether the refreshing aroma of jasmine flower is obvious or not is the primary characteristic for consumers to evaluate its quality [3]. To analyze the roles of multiple rounds of scenting processes on jasmine tea aroma quality, the tea samples were subjected to aroma evaluation by a well-trained sensory panel. As shown in Figure 2, the aroma freshness, concentration, purity, and persistence of jasmine tea gradually intensified as the number of scenting processes increased. This demonstrated that repeated scenting processes are necessary to elevate the aroma quality of jasmine tea. Notably, all four aroma attributes were significantly improved from the BT to R2 samples, with smaller but still significant improvements in the R2 to R4 samples. Moreover, compared with the R4 sample, the Tihua process played a significantly role in improving the refreshing aroma of the T1 sample, but the effect on the concentration, persistence, and purity of jasmine aroma was not significant (Appendix A). This may be due to the smaller amount of flowers used in this process and the short scenting time. Considering this process without heat-drying might help some of the low boiling point aroma substances remain in the tea leaves, thereby enhancing the refreshing aroma of jasmine tea. Taken together, the scenting processes (especially the first two rounds) significantly improved the aroma freshness, concentration, purity, and persistence of jasmine tea, while the Tihua processes mainly enhanced its refreshing aroma.

### 3.2. Overview of VOC Profiling

In this study, to comprehensively analyze the metabolite basis of jasmine tea aroma formation, the WTV method-based volatilomics was employed. A total of 887 VOCs was identified for all the samples (Appendix A). To the best of our knowledge, we identified the largest number of VOCs in both baked green tea and jasmine tea compared to previous studies [2,3,5,6,7,25,26,27]. In detail, BT, R1, and R2 samples possessed 667, 828, and 883 VOCs, respectively, while 887 VOCs were included in R3, R4, and T1, and this was due to the high sensitivity and volatile annotation brought by the WTV method. We calculated the relative content of total VOCs as the sum of the individual VOC [28]. The total amounts of VOCs corresponding to the samples were 27,435.56, 185,749.22, 761,973.14, 995,485.90, 1,051,196.48, and 1,103,055.54 μg/kg, respectively. It was observed that the total amount of VOCs increased with the number of scenting process, and especially the total amount of VOCs increased significantly in the R1 and R2 samples relative to the BT sample. At the same time, the number of VOCs in R1 and R2 samples was significantly higher than that of the BT sample, but it did not increase in jasmine teas (R3, R4, and T1) after two rounds of scenting (Figure 3A). This may be due to the increase of scenting times; the VOCs accumulated in tea leaves gradually tended to saturate. Based on their chemical structures, these VOCs can be classified into 16 categories, including esters, terpenoids, heterocyclic compounds, hydrocarbons, alcohols, ketones, aromatics, aldehydes, nitrogen compounds, acids, halogenated hydrocarbons, sulfur compounds, phenols, amines, ethers, and others. The proportions of esters and terpenoids were much higher in tea with the scenting process than in the BT sample (Figure 3B).

To assess the credibility of the detection results, we overlaid the total ion flow (TIC) diagram of the QC samples, which showed a good overlap (Appendix A). At the same time, the analysis of the metabolite CV values revealed that the percentage of metabolites with CV values less than 0.5 in QC samples was higher than 85%, and the percentage of metabolites with CV values less than 0.3 in QC samples was higher than 75% (Appendix A). These indicated that the stability of the HS-SPME and GC–MS instruments was good, which guaranteed the reliability of the detection results. To analyze the repeatability between intra-group or inter-group samples, the PCC analysis was performed. The result showed that three replicate samples of each group had better reproducibility (the r values were close to 1) (Appendix A). Moreover, the r values among R3, R4, and T1 samples were all close to 1, indicating that the VOCs of jasmine tea during multiple rounds of scenting processes were positively correlated. To further visualize the overview of the difference in the VOC phenotypes, unsupervised PCA was carried out. As shown in Figure 3C, the inter-group samples were distinguished from one another, and the BT sample and samples from different rounds of scenting processes were generally separated along the PC1. The first to five principal components (PC1 to PC5) explaining 72.76%, 11.18%, 4.74%, 2.81%, and 1.99% of the variations, respectively, and the total cumulative portion of the top five principal components was 93.49% (Figure 3D), indicating that the PCA yielded sufficient results for the repeated scenting processes of jasmine tea. This observation manifested that there was significant diversity of the VOCs from the BT sample and the tea samples with different scenting processes. In addition, we found that the sample clusters of R3, R4, and T1 were distributed relatively tighter than the R1 and R2 samples, which is consistent with the change trends of both the amount and number of total VOCs among jasmine tea samples (Figure 3A). 

### 3.3. Screening for Differential VOCs

To refine the statistical analysis for dramatically changed VOCs, OPLS-DA was used for pairwise comparisons between BT and jasmine tea with different rounds of scenting processes (R1 vs. BT, R2 vs. BT, R3 vs. BT, R4 vs. BT, and R4 vs. T1). In these five OPLS-DA models, samples were effectively distinguished from one another (Appendix A). The R^2^ Y and Q^2^ scores were higher than 0.9, meaning that the model explained variance and was highly predictive. Furthermore, the permutation test results showed that the *P* values of Perm R^2^ Y and Perm Q^2^ of all these five models were <0.005, demonstrating that the models were without overfitting. One the basis of the OPLS-DA, we identified a total of 664 differential VOCs in the above-mentioned pairwise comparisons (Appendix A). The Venn diagram (Figure 4A) showed that they shared 427 common differential VOCs and have nine, four, three, five, and ten distinctive differential VOCs, respectively. Moreover, the number of significantly differential metabolites showed that upregulated metabolites much exceeded the downregulated metabolites in all of the pairwise comparisons (Appendix A). The downregulated VOCs might be due to additional rounds of heat treatment for scented tea manufacturing, and some low boiling volatiles were heat-desorbed [12]. The number of upregulated compounds increased significantly with the increase in the number of scenting processes. Figure 4B shows the chemical classification of differential metabolites within the four pairwise comparisons. The terpinoids and esters were the most abundant differential VOCs. Previous studies have reported that esters such as benzyl acetate, methyl salicylate, methyl anthranilate, and 3-hexen-1-ol benzoate, and terpenoids such as α-farnesene, linalool, and geraniol are the key odorants of jasmine flowers [9,15], suggesting that the odorants of jasmine tea were largely absorbed from jasmine flowers. We also compared the metabolite differences between consecutive steps (R2 vs. R1, R3 vs. R2, R4 vs. R3, and T1 vs. R4) by OPLS-DA. Expectedly, samples were effectively distinguished from one another, and satisfied the R^2^ Y and Q^2^ scores > 0.9, and *P* values of Perm R^2^ Y and Perm Q^2^ were < 0.005 (Appendix A). A total of 497 differential VOCs was in these four compared combinations, among which 478, 69, 4, and 8 differential VOCs belong to R2 vs. R1, R3 vs. R2, R4 vs. R3, and T1 vs. R4 comparisons, respectively. In addition, there were no common differential VOCs shared by these four pairwise comparisons (Figure 4C). This result further explain that the first two rounds of scenting had a great influence on the aroma components of jasmine tea. Moreover, the number of significantly differential metabolites showed that upregulated metabolites much exceeded the downregulated metabolites in all of the pairwise comparisons (Appendix A). Similarly, terpenoids and esters were the most abundant differential VOCs (Figure 4D).

### 3.4. Dynamic Changes of the Different VOCs of Jasmine Tea during Multiple Rounds of Scenting

To investigate the dynamic changes of differential VOCs that occurred during the steps of repeated scenting, K-means clustering was applied. Five different variation tendencies were defined among the differential VOCs (Figure 5). Previous research has shown that the base tea has different absorption efficiency for different volatiles [29]. From the results, we could determine the key odorants associated with the aroma basis of our processed jasmine teas.

#### 3.4.1. Dynamic Changes of the Different VOCs in Subclass 1

There were 283 differential VOCs occupying the largest number of differential VOCs, which were clustered into subclass 1, which showed an upward trend from BT to R4 but kept relatively steady from R4 to T1 (Figure 5A). This change trend was consistent with the sensory evaluation about aroma concentration, purity, and persistence of jasmine teas (Appendix A), implying that these chemicals were important for the aroma formation of jasmine tea during multiple rounds of scenting. In this class, the content of esters was the most abundant (30.29%), followed by terpenoids (23.34%), heterocyclic compounds (16.62%), alcohols (8.84%), etc. (Figure 5B). The relative content of these chemicals is listed in Appendix A, 58 out of which were over 1000 μg/kg in jasmine teas. The contribution of VOCs to the aroma characteristics of jasmine tea can be determined not only by their contents but also on the odor threshold. We therefore calculated the OAVs for 50 VOCs in this class to evaluate the contribution of the different VOCs to jasmine tea, 34 out of which had OAVs > 1 in jasmine teas (Table 1). It was observed that methyl anthranilate was the one with the highest OAV (1728.31–14608.12) of the VOCs of importance to jasmine tea aroma, which was followed by (Z)-6-nonenal (1735.64–7906.54), methyl benzoate (50.76–1100.48), 2-nonenal (346.20–939.80), (Z)-3-hexen-1-ol acetate (240.27–776.60), methyl salicylate (138.93–751.13), linalool (123.73–666.06), (E,Z)-2,6-nonadienal (192.00–436.00), indole (19.01–349.44), 2-methoxy-phenol (130.71–340.47), hotrienol (52.06–298.85), 3-methylbutyl 3-methylbutanoate (20.77–107.06), α-farnesene (4.69–87.06), (E)-2-Hexenal (22.79–73.96), cis-3-hexenyl hexanoate (0.33–57.57), ethyl hexahydrobenzoate (14.00–56.33), octyl isobutyrate (3.29–41.44), cis-3-hexenyl butyrate (1.10–39.22), 2-methoxy-3-isopropylpyrazine (4.27–32.95), geranyl isobutyrate (1.55–32.31), (E,E)-2,4-decadienal (4.28–29.84), (6Z)-nonen-1-ol (17.86–25.20), (Z)-3-hexen-1-ol (3.51–12.48), fenchone (1.92–11.40), isomaltol (1.93–9.26), p-cresol (1.04–7.26), ethyl enanthate (2.50–6.27), geranyl acetate (2.89–3.79), ethyl (methylthio)acetate (1.01–3.43), 6-methylhept-5-en-2-one (0.42–2.01), isoterpinene (0.60–1.90), (Z)-2-decenal (0.68–1.61), etc. Compared to previous studies, our study detected the most abundant amount of VOCs. Most of these key odorants were already at their highest levels in R4, except for α-farnesene, methyl benzoate, methyl salicylate, (E,Z)-2,6-nonadienal, etc. This result was consistent with previous research in that the content of many key odorants did not increase significantly in the Tihua process [30]. However, Wang et al. found that single-petal jasmine had a higher α-farnesene content than that in double-petal jasmine and this may be an important reason for the fresh and elegant fragrance of single-petal jasmine [31]. In the present study, we found that the Tihua process mainly enhanced the refreshing aroma of jasmine tea, and the relative high content of α-farnesene in T1 samples may be an important cause. This suggests that we can adopt the strategy of using double-petal jasmine for scenting and single-petal jasmine for Tihua to improve the refreshing aroma quality of jasmine tea..

#### 3.4.2. Dynamic Changes of the Different VOCs in Subclass 2

Subclass 2 only had 28 different VOCs that showed upward and downward trends from BT to R1 and R1 to R3, respectively. Then there were no obvious changes from R3 to T1 (Figure 5C). This change trend was not consistent with our sensory evaluation results, suggesting that these VOCs were not important for the aroma quality of jasmine tea. Among these VOCs, the content of aromatics was over half (57.55%), followed by acids (12.74%), terpenoids (9.90%), alcohols (7.88%), etc. (Figure 5D). Only the content of 1,3-dimethyl-benzene was over 100 μg/kg in both BT and jasmine tea samples (Appendix A). Only 1,3,8-P-menthatriene, 1-octene, and 3-methoxy-2,5-dimethylpyrazine had OAVs > 1 in BT or jasmine teas due to their relative low odor threshold (Table 1). These VOCs were not unique to jasmine flowers.

#### 3.4.3. Dynamic Changes of the Different VOCs in Subclass 3

There were 68 VOCs of subclass 3 from the K-means clustering analysis that showed an upward trend from BT to R2 but dropped slightly from R2 to T1 (Figure 5E). Interestingly, the relative content of esters accounted for 96.96% (Figure 5F). We then specifically analyzed the content of the VOCs and found out this was due to the extremely high content of benzyl acetate, which reached 65623.35 μg/kg in the R3 sample (Appendix A). Benzyl acetate with high OAVs (115.11–243.05) was found in jasmine tea samples (Table 1). In addition to benzyl acetate, the OAVs of (E)-2-nonenal, benzaldehyde, and benzeneacetaldehyde were also higher than 1 in jasmine tea samples, indicating that these VOCs were key odorants for jasmine tea.

#### 3.4.4. Dynamic Changes of the Different VOCs in Subclass 4

In subclass 4, 263 differential VOCs showed upward trends from BT to T1, which was consistent with the results of a sensory evaluation of refreshing aroma (Figure 5G). Unlike the change trend of subclass 1, there was a significantly upward trend from R4 to T1 in subclass 4, suggesting that these VOCs were important for the refreshing aroma brought by the Tihua process. In this class, the content of esters (35.22%) and terpenoids (34.02%) was the most abundant (Figure 5H); the specific contents of these 263 VOCs are listed in Appendix A. There were 17 VOCs with OAVs > 1 in jasmine tea samples (Table 1), including 3-hexen-1-ol benzoate (3.31–276.19), benzyl alcohol (21.71–202.63), methyl enanthate (18.83–153.49), (E)-6-nonenal (33.43–77.14), ethylcinnamate (1.76–34.48), (E)-2-dodecenal (0.90–19.70), piperonal (0.58–15.73), benzoic acid (1.10–14.99), eugenol (0.33–12.37), nerolidol (0.15–11.79), 2-undecanone (0.18–5.30), ethyl salicylate (0.41–4.25), copaene (0.34–4.19), geraniol (0.37–3.35), (2-nitroethyl)-benzene (1.12–3.32), benzoic ether (0.15–2.22), benzyl benzoate (0.33–2.19), etc. These VOCs have been identified in jasmine flowers and are considered as key odorants for jasmine flowers [9,11,32]. Thus, these VOCs in jasmine tea were mainly absorbed from jasmine flowers, have high adsorption efficiency, and can also be effectively adsorbed in the Tihua process.

**Table 1 foods-12-00812-t001:** The odor activity values (OAVs) of differential volatile organic compounds.

No.	Volatile Compounds	Class	CAS	Odor Threshold (ug/kg)	OAV					
					BT	R1	R2	R3	R4	T1
	**Sub class 1**									
X1	2,3,5-Trimethylpyrazine	Heterocyclic compound	14667-55-1	10 (*)	-	11.85	35.39	34.65	38.17	34.71
X2	Isoterpinene	Terpenoids	586-62-9	260 (*)	0.24	0.60	1.24	1.80	1.85	1.90
X3	Fenchone	Terpenoids	1195-79-5	500 (*)	0.53	1.92	7.68	10.72	11.40	11.02
X4	Levomenthol	Terpenoids	2216-51-5	950 (*)	0.02	0.04	0.05	0.06	0.06	0.06
X5	α-Farnesene	Terpenoids	502-61-4	87 (a)	0.16	4.69	51.98	73.31	80.63	87.06
X6	Geranyl acetate	Terpenoids	105-87-3	100 (*)	-	-	2.89	3.27	3.70	3.79
X7	Linalool	Terpenoids	78-70-6	50 (*)	30.09	123.73	455.25	630.41	666.06	652.45
X8	p-Cresol	Phenol	106-44-5	2 (*)	-	1.04	4.20	5.97	7.21	7.26
X9	2-Methoxy-phenol	Phenol	90-05-1	0.17 (*)	61.76	130.71	241.06	314.12	337.18	340.47
X10	4-Ethyl-2-methoxy-phenol	Phenol	2785-89-9	25 (*)	-	0.12	0.25	0.24	0.25	0.25
X11	Methyleugenol	Phenol	93-15-2	68 (*)	-	-	0.04	0.06	0.07	0.08
X12	6-Methylhept-5-en-2-one	Ketone	110-93-0	100 (*)	0.09	0.42	1.62	1.90	2.01	1.99
X13	2-Methoxy-3-isopropylpyrazine	Heterocyclic compound	25773-40-4	20 (*)	0.99	4.27	20.18	31.25	32.95	31.63
X14	Isomaltol	Heterocyclic compound	3420-59-5	7.5 (*)	0.38	1.93	7.18	8.55	9.26	9.05
X15	3,5-Dimethyl-2-ethylpyrazine	Heterocyclic compound	13925-07-0	5	0.24	0.46	0.68	0.85	0.82	0.90
X16	Indole	Heterocyclic compound	120-72-9	140 (b)	-	19.01	259.66	338.69	349.44	347.98
X17	Heptyl acetate	Ester	112-06-1	90 (*)	0.02	0.03	0.05	0.07	0.09	0.07
X18	cis-3-Hexenyl butyrate	Ester	16491-36-4	32 (*)	0.26	1.10	39.22	34.46	35.19	33.59
X19	Ethyl (methylthio)acetate	Ester	4455-13-4	25 (*)	-	1.01	3.19	3.11	3.43	3.15
X20	Methyl benzoate	Ester	93-58-3	28 (*)	11.71	50.76	894.01	969.62	1084.96	1100.48
X21	3-Methylbutyl 3-methylbutanoate	Ester	659-70-1	20 (*)	7.81	20.77	72.98	101.13	107.06	104.13
X22	(Z)-3-hexen-1-ol acetate	Ester	3681-71-8	13 (*)	1.67	240.27	727.61	710.26	776.60	700.25
X23	Ethyl enanthate	Ester	106-30-9	2 (*)	1.81	2.50	5.00	6.01	6.27	6.03
X24	Ethyl hexahydrobenzoate	Ester	3289-28-9	0.06 (*)	7.00	14.00	48.00	39.00	45.33	56.33
X25	Methyl salicylate	Ester	119-36-8	40 (c)	6.28	138.93	496.61	739.84	748.85	751.13
X26	Methyl 2-(methylamino)benzoate	Ester	85-91-6	20.3 (d)	-	9.67	55.22	79.60	82.95	83.96
X27	cis-3-Hexenyl hexanoate	Ester	31501-11-8	16 (e)	-	0.33	44.00	49.49	55.76	57.57
X28	Octyl isobutyrate	Ester	109-15-9	6 (*)	-	3.29	24.89	36.70	40.78	41.44
X29	N-Decyl ethanoate	Ester	112-17-4	40 (*)	0.10	0.14	0.25	0.28	0.30	0.32
X30	Geranyl isobutyrate	Ester	2345-26-8	450 (*)	0.04	1.55	18.88	26.95	29.92	32.31
X31	Methyl anthranilate	Ester	134-20-3	3 (d)	-	1728.31	11443.59	14252.70	14608.12	14252.21
X32	(Z)-2-Decenal	Aldehyde	2497-25-8	100 (*)	0.43	0.68	1.19	1.31	1.49	1.61
X33	(E)-2-Hexenal	Aldehyde	6728-26-3	17 (f)	-	22.79	73.96	72.31	70.32	63.58
X34	(Z)-6-Nonenal	Aldehyde	2277-19-2	1 (*)	540.24	1735.64	5512.78	7485.60	7906.54	7634.56
X35	2-Nonenal	Aldehyde	2463-53-8	0.5 (*)	58.88	346.20	836.80	939.80	918.20	916.40
X36	(E,Z)-2,6-Nonadienal	Aldehyde	557-48-2	0.01 (*)	108.00	192.00	296.00	360.00	420.00	436.00
X37	(E,E)-2,4-Decadienal	Aldehyde	25152-84-5	0.5 (*)	0.96	4.28	25.88	26.92	29.84	29.48
X38	Hotrienol	Alcohol	20053-88-7	110 (*)	11.85	52.06	203.28	282.91	298.85	293.09
X39	(Z)-3-Hexen-1-ol	Alcohol	928-96-1	200 (*)	-	3.51	12.74	12.27	12.48	11.81
X40	2-Ethyl-1-hexanol	Alcohol	104-76-7	270 (*)	-	0.01	0.02	0.02	0.02	0.02
X41	(6Z)-Nonen-1-ol	Alcohol	35854-86-5	1 (*)	9.04	17.86	20.78	24.80	25.20	24.54
X42	P-Hydroxybenzoic acid	Acid	99-96-7	40,000 (*)	-	-	0.08	0.12	0.13	0.15
	**Sub class 2**									
X43	(Z)-2-Penten-1-ol	Alcohol	1576-95-0	720 (*)	-	0.29	-	-	-	-
X44	3-Carene	Terpenoids	13466-78-9	37 (*)	0.28	0.39	0.54	0.40	0.33	0.27
X45	1,3,8-P-Menthatriene	Terpenoids	18368-95-1	15 (*)	2.22	3.24	3.43	0.92	1.09	0.91
X46	1-Octene	Hydrocarbons	111-66-0	0.5 (*)	29.36	35.64	14.88	11.00	14.12	11.84
X47	3-Methoxy-2,5-dimethylpyrazine	Heterocyclic compound	19846-22-1	0.1 (*)	16.40	38.60	32.20	26.80	28.80	27.20
X48	Methyl caproate	Ester	106-70-7	70 (*)	-	0.05	0.01	0.01	0.01	0.01
X49	Naphthalene	Aromatics	91-20-3	500 (*)	0.23	0.15	0.11	0.06	0.06	0.05
X50	(1-Methylethyl)-benzene	Aromatics	98-82-8	70 (*)	-	0.02	0.02	-	0.01	0.01
X51	Trans-Anethole	Aromatics	4180-23-8	15 (*)	0.00	0.51	0.34	0.21	0.18	0.24
	**Sub class 3**									
X52	cis-Ocimene	Terpenoids	3338-55-4	55 (*)	0.63	2.01	6.61	4.41	4.03	3.55
X53	β-Ionone	Terpenoids	14901-07-6	3.5 (*)	20.42	37.95	55.00	49.59	55.94	58.02
X54	β-Myrcene	Terpenoids	123-35-3	42 (e)	0.43	0.57	1.08	0.76	0.69	0.55
X55	2,2-Dimethyl-3-hexanone	Ketone	5405-79-8	0.82 (*)	-	51.05	72.51	53.22	73.59	54.51
X56	2,2,6-Trimethylcyclohexan-1-one	Ketone	2408-37-9	100 (*)	0.14	0.38	0.41	0.36	0.42	0.40
X57	1-Hepten-3-one	Ketone	2918-13-0	0.04 (*)	-	25.50	23.50	24.00	18.50	16.50
X58	Isopentyl acetate	Ester	123-92-2	0.15 (*)	-	53.20	92.00	54.40	44.93	40.80
X59	Amyl acetate	Ester	628-63-7	43 (*)	-	0.11	0.16	0.14	0.13	0.11
X60	5-Decanolide	Ester	705-86-2	66 (*)	0.05	0.10	0.12	0.11	0.12	0.14
X61	Geranyl formate	Ester	105-86-2	200 (*)	0.05	0.08	0.12	0.11	0.13	0.12
X62	Octyl butyrate	Ester	110-39-4	250 (*)	0.20	0.50	0.59	0.55	0.63	0.56
X63	Benzyl acetate	Ester	140-11-4	270 (d)	4.09	115.11	223.57	243.05	234.82	228.95
X64	(E)-2-Nonenal	Aldehyde	18829-56-6	0.19 (*)	-	128.11	133.16	152.63	164.00	184.11
X65	Benzaldehyde	Aldehyde	100-52-7	3 (g)	19.81	47.99	55.44	54.85	52.91	51.91
X66	Benzeneacetaldehyde	Aldehyde	122-78-1	4 (g)	10.91	15.81	47.97	36.79	30.22	35.20
X67	Undecanal	Aldehyde	112-44-7	12.5 (*)	0.09	0.16	0.26	0.22	0.25	0.28
X68	1-Hexanol	Alcohol	111-27-3	5.6 (*)	0.00	4.06	5.25	5.40	4.67	5.33
	**Sub class 4**									
X69	Copaene	Terpenoids	3856-25-5	140 (*)	0.03	0.34	1.68	2.95	3.86	4.19
X70	Selina-4(15),7(11)-diene	Terpenoids	515-17-3	2600 (*)	-	-	0.02	0.04	0.04	0.05
X71	Geraniol	Terpenoids	106-24-1	75 (e)	0.04	0.37	1.47	2.14	2.56	3.35
X72	Nerolidol	Terpenoids	40716-66-3	10 (a)	0.10	0.15	2.43	4.77	8.32	11.79
X73	Eugenol	Phenol	97-53-0	90 (*)	-	0.33	3.68	8.06	10.29	12.37
X74	2-Undecanone	Ketone	112-12-9	450 (*)	-	0.18	2.58	4.59	5.30	5.23
X75	Benzoic ether	Ester	93-89-0	575 (*)	-	0.15	1.06	1.50	1.95	2.22
X76	Methyl enanthate	Ester	106-73-0	4 (*)	-	18.83	63.61	121.48	142.42	153.49
X77	Isopentyl hexanoate	Ester	2198-61-0	900 (*)	-	0.01	0.03	0.03	0.04	0.04
X78	Ethyl salicylate	Ester	118-61-6	84 (*)	-	0.41	1.39	3.29	3.72	4.25
X79	Ethylcinnamate	Ester	4192-77-2	1 (*)	-	1.76	11.96	23.80	29.90	34.48
X80	Ethyl pelargonate	Ester	123-29-5	1200 (*)	-	0.03	0.47	0.87	0.99	0.99
X81	Hexyl caproate	Ester	6378-65-0	6400 (*)	-	-	0.00	0.00	0.01	0.01
X82	Ethylcinnamate	Ester	103-36-6	40 (*)	-	0.03	0.26	0.52	0.67	0.77
X83	Benzyl benzoate	Ester	120-51-4	341 (d)	-	-	0.33	1.03	1.95	2.19
X84	3-Hexen-1-ol benzoate	Ester	25152-85-6	110 (*)	-	3.31	129.72	229.91	271.66	276.12
X85	(2-Nitroethyl)-benzene	Aromatics	6125-24-2	2 (*)	-	-	1.12	1.80	2.61	3.32
X86	(E)-6-Nonenal	Aldehyde	2277-20-5	0.07 (*)	22.57	33.43	46.86	55.14	77.14	74.86
X87	(E)-2-Dodecenal	Aldehyde	20407-84-5	1.4 (*)	0.33	0.90	6.50	13.20	17.14	19.70
X88	Piperonal	Aldehyde	120-57-0	3.9 (*)	-	0.58	7.27	12.29	14.57	15.73
X89	(2E,4Z)-2,4-Decadienal	Aldehyde	25152-83-4	4.2 (*)	0.04	0.05	0.14	0.18	0.26	0.22
X90	Benzyl alcohol	Alcohol	100-51-6	100 (h)	0.22	21.71	82.16	158.27	186.52	202.63
X91	Benzoic acid	Acid	65-85-0	85 (*)	0.08	1.10	7.16	10.10	13.22	14.99
	**Sub class 5**									
X92	3-Ethyl-2,5-dimethylpyrazine	Heterocyclic compound	13360-65-1	25 (*)	0.05	0.12	0.11	0.09	0.11	0.17

*: Compilations of odor threshold values in air, water, and other media. Edition 2011./L.J.van Gemert; a: [33]; b: [34]; c: [35]; d: [36]; e: [37]; f: [38]; g: [39]; h: [40].

#### 3.4.5. Dynamic Changes of the Different VOCs in Subclass 5

The change tendency of the 28 differential VOCs in subclass 5 was a slight increase from BT to R4, followed by a significant increase from R4 to T1 (Figure 5I). The heterocyclic compounds were the most abundant (45.66%). However, most of these VOCs were very low (Appendix A), and no VOCs in jasmine tea samples had an OAV value greater than 1 (Table 1). In addition, there are no reports about these VOCs in jasmine flowers. Thus, these VOCs may not be important for the aroma of jasmine tea.

### 3.5. Correlation Analysis between Refreshing Aroma and Characteristic VOCs of Jasmine Tea

To further confirm the relationships between the VOCs and the refreshing aroma, PLSR analysis was applied to correlate the key volatiles (OAVs ≥ 1 and VIP > 1) with the refreshing aroma. PLSR is a linear regression model that simultaneously projects the independent variables (X) and dependent variables (Y) to a new space with the constraint that the components can explain as much as possible the variance between X and Y so that it not only can extract useful information as much as possible but also strengthen the correlation between X and Y [41]. In total, 65 VOCs with OAV > 1, as shown in Table 1, were used as X variables, and freshness scores were used as Y variables, and then PLSR was used to analyze the VOCs and freshness scores of the samples.

As shown in Figure 6, except for β-myrcene (X54), which lay between the two ellipses representing 50% and 75% of the explained variance, the other 64 VOCs were located between the two ellipses that represented the explained variance of 75% and 100%, which means that these variates could be well explained by this model, and that these 64 VOCs constitute the freshness characteristic of the samples. Freshness is located in the first quadrant, close to ethyl (methylthio)acetate (X19, green and fruity odor), (Z)-3-hexen-1-ol acetate (X22, fresh, green and sweet odor), (E)-2-hexenal (X33, green odor), 2-nonenal (X35, green odor), (Z)-3-hexen-1-ol (X39, fresh and green odor), (6Z)-nonen-1-ol (X41, fresh and green odor), β-ionone (X53, floral, sweet and fruity odor), and benzyl acetate (X63, fruity, jasmine, and fresh odor). What is more, the OAV values of ethyl (methylthio)acetate, (E)-2-hexenal, and (Z)-3-hexen-1-ol were greater than 1 in jasmine tea samples, but less than 1 in BT, and OAVs of the remaining five differential VOCs in all samples were greater than 1 (Table 1). Therefore, these eight VOCs were the key odorants responsible for the refreshing aroma of jasmine tea. Methyl benzoate (X20, green and floral odor), benzyl alcohol (X90, floral and rose odor), α-farnesene (X5, green odor), linalool (X7, floral, sweet and green odor), indole (X16, floral odor), and methyl anthranilate (X31, fruity and flower odor), etc., were located in the positive direction of the x-axis and contributed to the enhancement of freshness, similar to previous studies [2,5,6,42]. Moreover, 1,3,8-P-menthatriene (X45, turpentine odor) and 1-octene (X46, gasoline odor) were located in the negative direction of the X-axis, indicating that freshness was negatively correlated with turpentine and gasoline odors. Benzyl acetate, benzyl alcohol, α-farnesene, methyl benzoate, methyl anthranilate, and linalool, etc., were identified as the main volatile compounds of jasmine tea [6,43]. The increase in the relative content of VOCs and the odor contribute to the freshness of the samples [19]. The relative content of these VOCs in jasmine tea samples was increased with the number of scenting processes; for example, the relative content of benzyl acetate, (E)-2-hexenal, and α-farnesene increased from 1104.78 ± 94.67 (BT), 0 (BT), and 13.58 ± 4.69 (BT) to 61816.03 ± 2105.26 (T1), 1080.78 ± 39.13 (T1), and 7574.61 ± 377.8 (T1), respectively. Likewise, the odor and fragrance of tea are correlated to the odor of volatile components that contribute to the various aroma characteristics of tea. For instance, benzyl acetate is a key odorant of jasmine flower and is a fragrance ingredient used in many compounds [44]. α-Farnesene, presenting a floral odor, was one of the major volatile constituents of jasmine flowers [45]. Methyl benzoate has a pleasant fruity odor and is the most abundant scenting compound in the majority of snapdragon varieties [46]. (Z)-3-Hexen-1-ol is a significantly important active component that contributes to the “green, grass, and fresh” odor in vegetable food, such as grapes, passion fruits, and *Toona sinensis* (A. Juss.) Roem [42,47,48]. Moreover, it has been regarded as the main source of green odor in green tea. The odor–taste interaction of geraniol or β-ionone could enhance the sweet odor, which may help with the freshness of the jasmine tea [49].

From the above results, it can be seen that most of the VOCs with OAV > 1 contributed positively to freshness, except for 1,3,8-P-menthatriene and 1-octene. Ethyl (methylthio) acetate, (Z)-3-hexen-1-ol acetate, (E)-2-hexenal, 2-nonenal, (Z)-3-hexen-1-ol, (6Z)-nonen-1-ol, β-ionone, and benzyl acetate were identified as the key odorants responsible for the freshness aroma of jasmine tea.

## 4. Conclusions

In the present study, jasmine teas with different rounds of scenting processes were produced to analyze the key odorants responsible for jasmine tea aroma quality. The results of the sensory evaluation showed that the scenting processes (especially the first two rounds) significantly improved the aroma freshness, concentration, purity, and persistence of jasmine tea, while the Tihua processes mainly enhanced the refreshing aroma of jasmine tea. Using GC–MS-based WTV method, we identified 887 VOCs among the base tea and jasmine tea samples; these VOCs can be classified into 16 categories, including esters, terpenoids, heterocyclic compounds, hydrocarbons, alcohols, ketones, aromatics, aldehydes, nitrogen compounds, acids, halogenated hydrocarbons, sulfur compounds, phenols, amines, ethers, and others. Combined with multivariate statistical analysis methods and OAVs, we identified that ethyl (methylthio)acetate, (Z)-3-hexen-1-ol acetate, (E)-2-hexenal, 2-nonenal, (Z)-3-hexen-1-ol, (6Z)-nonen-1-ol, β-ionone, and benzyl acetate were key odorants responsible for the freshness aroma of jasmine tea. Further precise quantification as well as sensory and biological studies are needed to ascertain the effects of each key odorant on the aroma and health benefits of jasmine tea.

## Figures and Tables

**Figure 1 foods-12-00812-f001:**
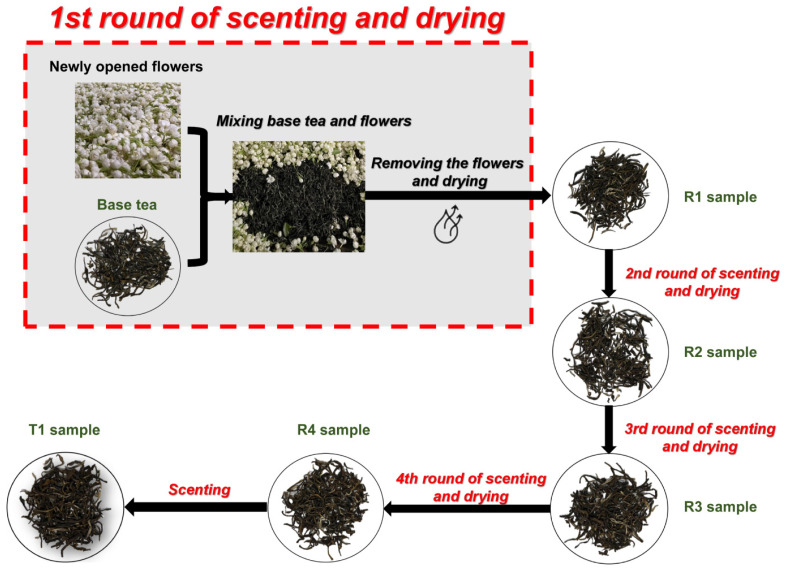
Schematic diagram of the manufacturing process of jasmine tea. The “Scenting” for the T1 sample in the diagram means “Tihua” in this article.

**Figure 2 foods-12-00812-f002:**
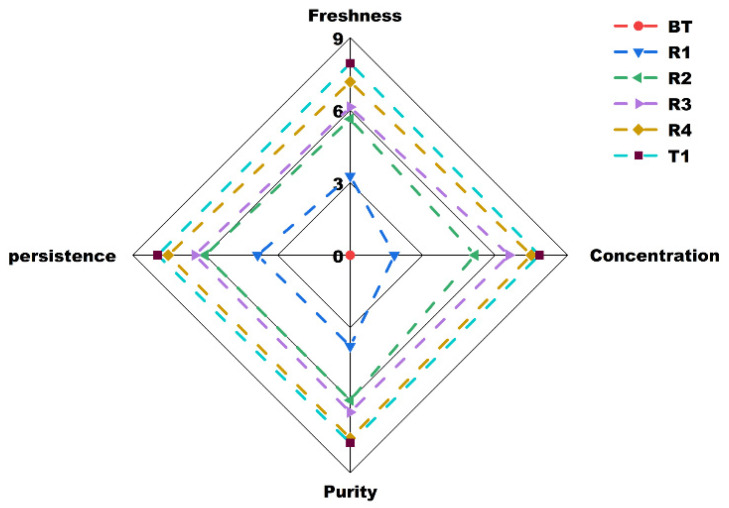
Spider plot of the aroma profiles of baked green tea (used as base tea) and jasmine tea with different rounds of scenting processes. BT: base tea; R1, R2, R3, R4: jasmine tea with one (R1), two (R2), three (R3), and four (R4) rounds of scenting processes; T1: jasmine tea with four rounds of scenting processes and Tihua process.

**Figure 3 foods-12-00812-f003:**
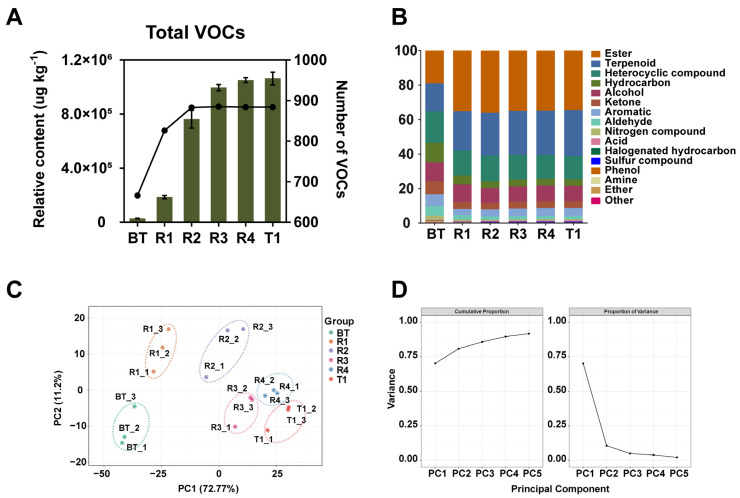
Overview of the profiling of volatile organic compounds (VOCs) in base tea and tea with different rounds of scenting. (**A**) Dynamic changes of the relative content and number of total VOCs in base tea and tea with different rounds of scenting. The column and line represent the relative content and number of VOCs, respectively. (**B**) Classification of the identified VOCs in samples of base tea and tea with different rounds of the scenting process based on their chemical structures and relative contents. (**C**) The principal component analysis (PCA) score plot of samples of base tea and tea with different rounds of the scenting process. (**D**) The cumulative proportion and proportion of variance for the top five principal components from the PCA model. BT: base tea; R1, R2, R3, R4: jasmine tea with one (R1), two (R2), three (R3), and four (R4) rounds of scenting processes; T1: jasmine tea with four rounds of scenting processes and the Tihua process.

**Figure 4 foods-12-00812-f004:**
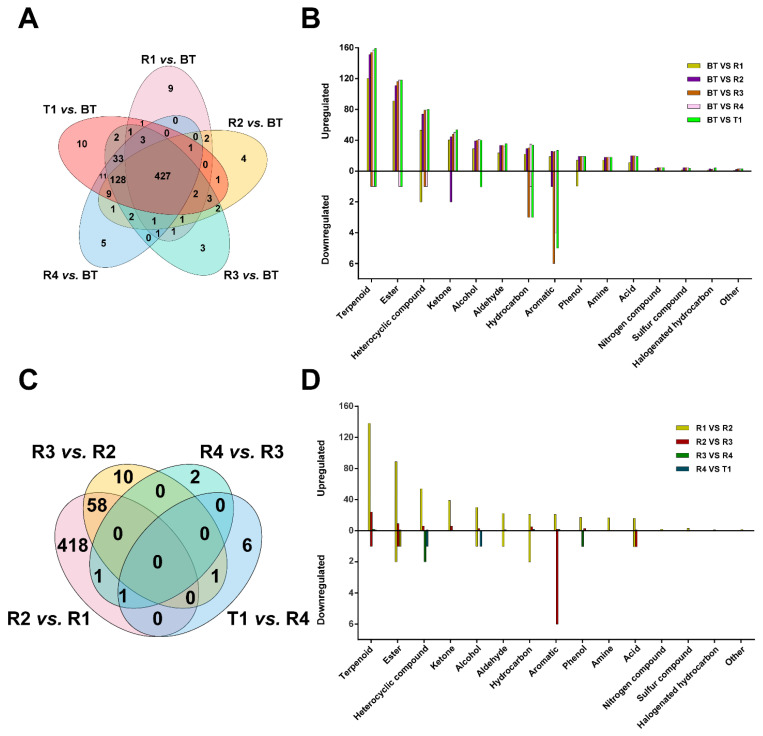
Differential volatile organic compounds (VOCs) in the six tea samples. (**A**) Venn diagram of the differential VOCs of R1 vs. BT, R2 vs. BT, R3 vs. BT, R4 vs. BT, T1 vs. BT; (**B**) Chemical classification of differential VOCs within the five pairwise comparisons; (**C**) Venn diagram of the differential VOCs of R2 vs. R1, R3 vs. R2, R4 vs. R3, T1 vs. R4; (**D**) Chemical classification of differential VOCs within the four pairwise comparisons. BT: base tea; R1, R2, R3, R4: jasmine tea with one (R1), two (R2), three (R3), and four (R4) rounds of scenting processes; T1: jasmine tea with four rounds of scenting processes and the Tihua process.

**Figure 5 foods-12-00812-f005:**
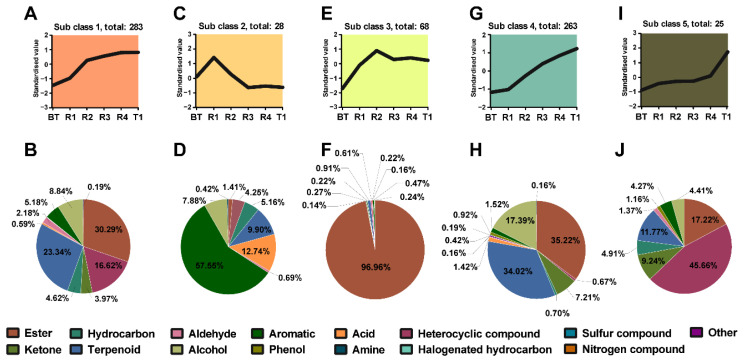
K-means clustering of differential volatile organic compounds (VOCs) in Subclass 1 (**A**), Subclass 2 (**C**), Subclass 3 (**E**), Subclass 4 (**G**), and Subclass 5 (**I**). Chemical classification of the differential VOCs in Sub class 1 (**B**), Sub class 2 (**D**), Sub class 3 (**F**), Sub class 4 (**H**), and Sub class 5 (**J**). BT: base tea; R1, R2, R3, R4: jasmine tea with one (R1), two (R2), three (R3), and four (R4) rounds of scenting processes; T1: jasmine tea with four rounds of scenting processes and the Tihua process.

**Figure 6 foods-12-00812-f006:**
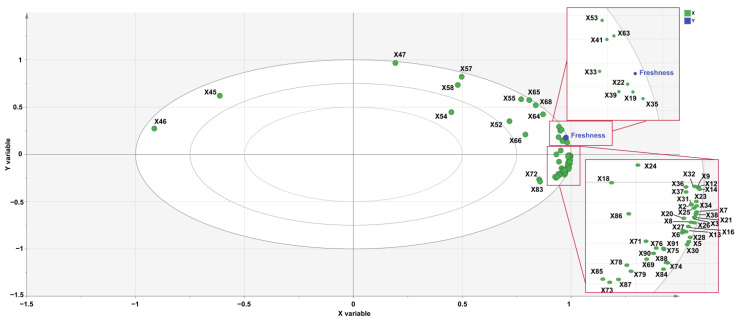
Correlation loading plots analyzed by PLSR. Variant X and Y represent VOCs with OAV > 1 and score of freshness profile evaluation. Green dots represent VOCs with OAV > 1, and blue dots represent freshness. Ellipses represent r^2^ = 0.5, 0.75, and 1.0, respectively.

## Data Availability

All of the data included in this study are available upon request by contacting the corresponding author.

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
