# Peer review of "Volatilomics Analysis of Jasmine Tea during Multiple Rounds of Scenting Processes"

_foods, 2023, doi:10.3390/foods12040812_

Round 1
Reviewer 1 Report
The purpose of the study, the methods used and the statistical evaluation contain sufficient information. The results have been also discussed and explained in detail.
It can be accepted in its current state.
Author Response
Response to Reviewer 1 Comments
Point 1: The purpose of the study, the methods used and the statistical evaluation contain sufficient information. The results have been also discussed and explained in detail.
It can be accepted in its current state.
Response 1: Thank you for taking the time to evaluate our manuscript! We will further improve the content of the manuscript according to the requirements of the foods journal before it is accepted.
Reviewer 2 Report
Manuscript is suitable for this Journal. The manuscript "Widely Targeted Volatilomics Analysis Reveals Dynamic Changes in Volatile Organic Compounds During Multiple Rounds of Scenting Processes of Jasmine Tea" is well written, the idea of this paper is very interesting. The title of the manuscript is suitable of researching. Aim of the paper is clear. Introduction, Materials and Methods and Results and Discussion are well written. The conclusion is connected with the obtained results and the contribution of the research is highlighted.
Author Response
Response to Reviewer 2 Comments
Point 1: Manuscript is suitable for this Journal. The manuscript "Widely Targeted Volatilomics Analysis Reveals Dynamic Changes in Volatile Organic Compounds During Multiple Rounds of Scenting Processes of Jasmine Tea" is well written, the idea of this paper is very interesting. The title of the manuscript is suitable of researching. Aim of the paper is clear. Introduction, Materials and Methods and Results and Discussion are well written. The conclusion is connected with the obtained results and the contribution of the research is highlighted.
Response 1: Thank you for taking the time to evaluate our manuscript! We will further improve the content of the manuscript according to the requirements of the foods journal before it is accepted.
Reviewer 3 Report
General comment
The manuscript ‘’Widely Targeted Volatilomics Analysis Reveals Dynamic Changes in Volatile Organic Compounds During Multiple Rounds of Scenting Processes of Jasmine Tea’’ described changes in Jasmine tea during the process of “scenting”. Results revealed that the aroma freshness, concentration, purity, and persistence of jasmine tea gradually intensifies as the number of scenting processes increases, and the last round of scenting process without drying plays a significant role in improving a refreshing aroma. A total of 887 volatile organic compounds were detected in jasmine tea samples. Among the great number of compounds ethyl (methylthio)acetate, (Z)-3-hexen-1-ol acetate, (E)-2-hexenal, 2-nonenal, (Z)-3-hexen-1-ol , (6Z)-nonen-1-ol, β-ionone, and benzyl acetate, were identified as key odorants responsible for the refreshing aroma of jasmine tea. Article is well done and it is worth to be published in Foods journal after minor corrections
Minor comments
Abstract
Line 17: add ‘‘(L.) Aiton‘‘ after ‘‘sambac‘‘
1. Introduction
Line 32: check the font of words ‘‘fragrance to‘‘
Lines 35, 37, 41, 44…: add space before bracket with references number through all manuscript (see one article published in journal Fodds or template)
Line 36: add ‘‘(L.) Aiton‘‘ after ‘‘sambac‘‘
2. Materials and Methods
Line 84: add ‘‘(L.) Kuntze‘‘ after ‘‘sinensis‘‘
Line 86: add abbreviation of authors name after ‘‘bifoliatum‘‘. I din’t find it as variety in Flora of China. Maybe it is a cultivar. Please, check.
Lines 88, 92, 95, 96…: remove upper symbol for the degree (like in line 135)
Line 174: something is probably missing before ‘‘smaller‘‘
3. Results and Discussion
Lines 220, 358: change ‘’Supplementary Figure S1’’ into ‘’Figure S1’’ (see foods-template)
Line 245: change ‘’Supplementary Table 1’’ into ‘’Table S1’’ (see foods-template)
Line 266: change ‘’Supplementary Figure S2’’ into ‘’Figure S2’’ (see foods-template)
Line 270: change ‘’Supplementary Figure S3’’ into ‘’Figure S3’’ (see foods-template)
Line 274 change ‘’Supplementary Figure S4’’ into ‘’Figure S4’’ (see foods-template)
Note below figures: note ‘’R1: jasmine tea with one round of scenting process; R2: jasmine tea with two rounds of scenting processes; R3: jasmine tea with three rounds of scenting processes; R4: jasmine tea with four rounds of scenting processes‘’ should be shortened as follows ‘’R1, R2, R3, R4: jasmine tea with one (R1), two (R2), three (R3), and four (R4) rounds of scenting processes;‘’
Line 305 change ‘’Supplementary Figure S5’’ into ‘’Figure S5’’
Line 315 change ‘’Supplementary Figure S6’’ into ‘’Figure S6’’
Line 328 change ‘’Supplementary Figure S7’’ into ‘’Figure S7’’
Lines 339-341, Figure 4: omit ‘’Figures should be placed in the main text near to the first time they are cited. A caption on a single line should be centered.‘’
Line 362: change ‘’Supplementary Table 2’’ into ‘’Table S2’’
Line 369: change ‘’Supplementary Table 3’’ into ‘’Table S3’’
Line 410: change ‘’Supplementary Table 4’’ into ‘’Table S4’’
Line 421: change ‘’Supplementary Table 5’’ into ‘’Table S5’’
Line 436: change ‘’Supplementary Table 6’’ into ‘’Table S6’’
Line 443: classificati on?
Table 1, X31, column R1, R2, R3: the column is too narrow for two decimal places. Think to use Landscape orientation for Table 1.
Table 1, X34, column R1: the same
Table 1, column CAS: the same
Line 491: (10.3390/molecules21030338)???
Line 497: add abbreviation of authors name after ‘‘sinensis‘‘
Line 508: add space before 1 at the beginning of row
References
Use article from journal Foods to see how to revised journal name in references list
Write title of articles on the same way. For example, see ref. no 25 and 26
Lines 564, 571: is it necessary dot after Molecules. See foods template
Line 568: journal abbreviation??
Line 616: delete no. 27
Lines 622, 624: journal abbreviations??
Line 627: put ‘‘Jasminum sambac‘‘ into Italic
Line 643: put ‘‘Camellia sinensis‘‘ into Italic
Line 662: put ‘‘Vitis vinifera‘‘ into Italic
Line 664: put ‘‘Passiflora edulis Sims f. flavicarpa‘‘
Author Response
Response to Reviewer 3 Comments
Point 1:
Minor comments
Abstract
Line 17: add ‘‘(L.) Aiton‘‘ after ‘‘sambac‘‘
Response 1: Thanks for pointing this out! We have added ‘‘(L.) Aiton‘‘ after ‘‘sambac‘‘ in revised manuscript.
Point 2:
- Introduction
Line 32: check the font of words ‘‘fragrance to‘‘
Lines 35, 37, 41, 44…: add space before bracket with references number through all manuscript (see one article published in journal Fodds or template)
Line 36: add ‘‘(L.) Aiton‘‘ after ‘‘sambac‘‘
Response 2: Thank you for pointing out these problems. We have corrected these mistakes in revised manuscript using track-changes.
Point 3:
- Materials and Methods
Line 84: add ‘‘(L.) Kuntze‘‘ after ‘‘sinensis‘‘
Line 86: add abbreviation of authors name after ‘‘bifoliatum‘‘. I din’t find it as variety in Flora of China. Maybe it is a cultivar. Please, check.
Lines 88, 92, 95, 96…: remove upper symbol for the degree (like in line 135)
Line 174: something is probably missing before ‘‘smaller‘‘
Response 3: Thank you for pointing out these problems. We have corrected these mistakes in revised manuscript using track-changes.
Point 4:
- Results and Discussion
Lines 220, 358: change ‘’Supplementary Figure S1’’ into ‘’Figure S1’’ (see foods-template)
Line 245: change ‘’Supplementary Table 1’’ into ‘’Table S1’’ (see foods-template)
Line 266: change ‘’Supplementary Figure S2’’ into ‘’Figure S2’’ (see foods-template)
Line 270: change ‘’Supplementary Figure S3’’ into ‘’Figure S3’’ (see foods-template)
Line 274 change ‘’Supplementary Figure S4’’ into ‘’Figure S4’’ (see foods-template)
Note below figures: note ‘’R1: jasmine tea with one round of scenting process; R2: jasmine tea with two rounds of scenting processes; R3: jasmine tea with three rounds of scenting processes; R4: jasmine tea with four rounds of scenting processes‘’ should be shortened as follows ‘’R1, R2, R3, R4: jasmine tea with one (R1), two (R2), three (R3), and four (R4) rounds of scenting processes;‘’
Line 305 change ‘’Supplementary Figure S5’’ into ‘’Figure S5’’
Line 315 change ‘’Supplementary Figure S6’’ into ‘’Figure S6’’
Line 328 change ‘’Supplementary Figure S7’’ into ‘’Figure S7’’
Lines 339-341, Figure 4: omit ‘’Figures should be placed in the main text near to the first time they are cited. A caption on a single line should be centered.‘’
Line 362: change ‘’Supplementary Table 2’’ into ‘’Table S2’’
Line 369: change ‘’Supplementary Table 3’’ into ‘’Table S3’’
Line 410: change ‘’Supplementary Table 4’’ into ‘’Table S4’’
Line 421: change ‘’Supplementary Table 5’’ into ‘’Table S5’’
Line 436: change ‘’Supplementary Table 6’’ into ‘’Table S6’’
Line 443: classificati on?
Table 1, X31, column R1, R2, R3: the column is too narrow for two decimal places. Think to use Landscape orientation for Table 1.
Table 1, X34, column R1: the same
Table 1, column CAS: the same
Line 491: (10.3390/molecules21030338)???
Line 497: add abbreviation of authors name after ‘‘sinensis‘‘
Line 508: add space before 1 at the beginning of row
Response 4: We are sorry for our careless mistakes. In revised manuscript, we have corrected all of the problems you raised using track-changes.
Point 5:
References
Use article from journal Foods to see how to revised journal name in references list
Write title of articles on the same way. For example, see ref. no 25 and 26
Lines 564, 571: is it necessary dot after Molecules. See foods template
Line 568: journal abbreviation??
Line 616: delete no. 27
Lines 622, 624: journal abbreviations??
Line 627: put ‘‘Jasminum sambac‘‘ into Italic
Line 643: put ‘‘Camellia sinensis‘‘ into Italic
Line 662: put ‘‘Vitis vinifera‘‘ into Italic
Line 664: put ‘‘Passiflora edulis Sims f. flavicarpa‘‘
Response 5: Thank you for pointing this out. In revised manuscript, we have corrected related problems you raised.
Reviewer 4 Report
Dear Authors,
In general, the manuscript is good to read, the structure of the work is clear and has a sufficient literature review. I present my comments below:
1. Title. The title seems a bit awkwardly worded, a bit too complicated. I suggest that the authors consider his change. For example: "Volatilomics Analysis of Jasmine Tea During Multiple Rounds of Scenting Processes".
2. Please check throughout the document for spaces before square brackets. In most cases, they are missing.
3. Lines 116-118. Are five panellists enough for this kind of sensory research? The age of the panellists can be entered.
4. Lines 234-244. This text is not the results of research. This information is more in line with the methodology.
5. Figure 3c. R1_1, R1_2, R1_3, R2_1, etc... too small, illegible font.
6. Line 491. No DOI needed.
Author Response
Response to Reviewer 4 Comments
Point 1: Title. The title seems a bit awkwardly worded, a bit too complicated. I suggest that the authors consider his change. For example: "Volatilomics Analysis of Jasmine Tea During Multiple Rounds of Scenting Processes".
Response 1: We are totally agree with your comment. In revised manuscript, we have changed the title to "Volatilomics Analysis of Jasmine Tea During Multiple Rounds of Scenting Processes".
Point 2: Please check throughout the document for spaces before square brackets. In most cases, they are missing.
Response 2: We are sorry for our mistakes. We have checked carefully and added spaces before square brackets in revised manuscript.
Point 3: Lines 116-118. Are five panellists enough for this kind of sensory research? The age of the panellists can be entered.
Response 3: Thank you for pointing this out. In fact, five-member panellists team was used in many studies on sensory evaluation of tea [1-5]. In revised manuscript, we have clearly indicated the age range of the panellists, details are as follow:
A panel of five expert panelists (two males, three females, and ranged from 40 to 53 years old) was recruited to evaluate the aroma profiles of different tea samples
References
- Liang, S.; Wang, F.; Granato, D.; Zhong, X.; Xiao, A.; Ye, Q.; Li, L.; Zou, C.; Yin, J.; Xu, Y.Effect of β-glucosidase on the aroma of liquid-fermented black tea juice as an ingredient for tea-based beverages.Food Chem. 2023, 402, 134201.
- Wang, J.; Shi, J.; Zhu, Y.; Ma, W.; Yan, H.; Shao, C.; Wang, M.; Zhang, Y.; Peng, Q.; Chen, Y., et al.Insights into crucial odourants dominating the characteristic flavour of citrus-white teas prepared from citrus reticulata Blanco 'Chachiensis' and Camellia sinensis'Fudingdabai'. Food Chem. 2022, 377, 132048.
- Deng, X.; Huang, G.; Tu, Q.; Zhou, H.; Li, Y.; Shi, H.; Wu, X.; Ren, H.; Huang, K.; He, X., et al.Evolution Analysis of Flavor-active Compounds During Artificial Fermentation of Pu-erh Tea.Food Chem. 2021,.
- Zhang, Q.; Hu, J.; Liu, M.; Shi, Y.; De Vos, R.C.H.; Ruan, J.Stimulated biosynthesis of delphinidin-related anthocyanins in tea shoots reducing the quality of green tea in summer.J Sci Food Agr. 2020, 100, 1505-1514.
- Ntezimana, B.; Li, Y.; He, C.; Yu, X.; Zhou, J.; Chen, Y.; Yu, Z.; Ni, D.Different Withering Times Affect Sensory Qualities, Chemical Components, and Nutritional Characteristics of Black Tea.Foods. 2021, 10, 2627.
Point 4: Lines 234-244. This text is not the results of research. This information is more in line with the methodology.
Response 4: Thank you for your valuable comment. We have removed the content of this part to “2.4. WTV Method” in revised manuscript.
Point 5: Figure 3c. R1_1, R1_2, R1_3, R2_1, etc... too small, illegible font.
Response 5: Thank you for pointing this out. We have enlarged the characters in the Figure 3c to identify them clearly.
Point 6: Line 491. No DOI needed.
Response 6: We are sorry for our careless mistakes. We have deleted DOI in line 491 of revised manuscript.